# CHOICES SPEAK LOUDER THAN QUESTIONS

**Gyeongje Cho**[*1]    **Yeonkyoung So**[*1]    **Jaejin Lee**[1,2]

[1]Graduate School of Data Science, Seoul National University
[2]Department of Computer Science, Seoul National University

gyeongje@aces.snu.ac.kr, kathy1028@snu.ac.kr, jaejin@snu.ac.kr

## ABSTRACT

Recent findings raise concerns about whether the evaluation of Multiple-Choice Question Answering (MCQA) accurately reflects the comprehension abilities of large language models. This paper explores the concept of *choice sensitivity*, which refers to the tendency for model decisions to be more influenced by the answer options than by a genuine understanding of the question. We introduce a new scoring method called *Normalized Probability Shift by the Question (NPSQ)*, designed to isolate the impact of the question itself and provide a more reliable assessment of comprehension. Through experiments involving various input formats, including cloze, symbols, and hybrid formats, we find that traditional scoring methods — such as those based on log-likelihood or its length-normalized variant — are vulnerable to superficial characteristics of the answer choices. In contrast, NPSQ remains stable even when modifications are made to the answer options.

## 1 INTRODUCTION

Multiple-choice question answering (MCQA) has emerged as a standard method for evaluating large language models (LLMs) due to its clear objectives, automatic grading, and alignment with human assessment protocols (Robinson et al., 2023; Achiam et al., 2023; Team et al., 2023; Jiang et al., 2024; Dubey et al., 2024). LLMs are tested across a variety of tasks within MCQA benchmarks (Clark et al., 2018; Talmor et al., 2019; Sakaguchi et al., 2021; Srivastava et al., 2023), ranging from commonsense reasoning (e.g., HellaSwag (Zellers et al., 2019)) to professional knowledge (e.g., MMLU (Hendrycks et al., 2020)). As LLMs advance rapidly, many have begun to achieve, and in some instances exceed, human-level performance on various MCQA benchmarks.

However, despite the widespread use of LLMs, concerns are growing regarding the reliability and fairness of their evaluation methods (Li et al., 2024; Lyu et al., 2024; Alzahrani et al., 2024; Molfese et al., 2025). Recent studies have demonstrated that even minor variations in prompt phrasing (Sclar et al., 2023; Zhuo et al., 2024; Zhu et al., 2024), the arrangement of few-shot examples (Zhao et al., 2021; Lu et al., 2022; Ma et al., 2023; Guo et al., 2024), or changes in the position of answer options (Zheng et al., 2023; Pezeshkpour & Hruschka, 2024; Wang et al., 2024) can significantly affect model performance. Observations indicate that LLMs often achieve high accuracy—higher than what might be expected from random guessing—when presented solely with answer choices (also referred to as options) without the corresponding questions (Balepur et al., 2024). This suggests that the final answer selected by a model may be influenced as much by external factors as by its comprehension of the question itself.

We would like to highlight a concerning observation: LLMs can sometimes arrive at correct answers by focusing solely on the answer choices, without even reading the question. This raises a fundamental issue about whether these models are genuinely understanding the task or merely exploiting patterns in the answer options. To illustrate this problem, consider a simple analogy: imagine a person who selects the correct answer to a multiple-choice question without reading the question itself, merely by looking at the options. Even if their choice is correct, it is difficult to argue that they truly understood the content or solved the problem as intended. The same reasoning applies to

---

[*]These authors contributed equally to this work.

LLMs. If a model can frequently choose the correct answer without relying on the actual question, its accuracy does not genuinely reflect true comprehension.

This paper examines the degree to which LLMs depend on answer choices instead of properly understanding the questions in MCQA benchmarks. We define and quantify a phenomenon called *choice sensitivity*, which occurs when model predictions are predominantly influenced by the provided answer choices rather than the actual questions. We empirically measure the prevalence of this phenomenon across various datasets, input formats, and model sizes.

Building on this analysis, we introduce a new evaluation method called *Normalized Probability Shift by the Question (NPSQ)*. This method more accurately isolates the impact of the question from that of the answer choices. Through extensive experiments, we demonstrate that traditional MCQA evaluation metrics are often highly sensitive to superficial features of the answer choices. In contrast, our proposed approach allows for a more robust and interpretable assessment of a model's true understanding of the questions.

## 2 RELATED WORK

MCQA has become a standard framework for evaluating LLMs. However, recent work has revealed that MCQA performance is highly sensitive to subtle changes in input formatting. Studies have shown that variations in prompt phrasing, the format of answer choices, and the order of examples in few-shot settings can lead to significant shifts in model predictions (Lu et al., 2022; Zheng et al., 2023; Pezeshkpour & Hruschka, 2024; Alzahrani et al., 2024).

These observations have brought into a question whether models truly comprehend the information provided in the prompt or are instead influenced by superficial aspects of the input. Consequently, many studies have sought to develop prompt selection and formatting techniques that more accurately capture a model's underlying language understanding capabilities (Webson & Pavlick, 2022; Wei et al., 2022; Min et al., 2022; Leidinger et al., 2023). Meanwhile, such findings raise serious concerns regarding the reliability and fairness of current evaluation methodologies (Li et al., 2024; Lyu et al., 2024; Wang et al., 2025; Balepur et al., 2025).

In addition to sensitivity to input formatting, another line of research has examined which components of the input most influence model decisions. Recent studies have investigated the true extent of model comprehension by evaluating performance under partial-input conditions, where only limited information from the prompt is provided (Gururangan et al., 2018; Poliak et al., 2018; Belinkov et al., 2019; Feng et al., 2019; Srikanth & Rudinger, 2022). These studies indicate that models can often solve problems or achieve high accuracy without needing the complete prompt. This suggests that correct answers can sometimes be reached without a true understanding of the problem.

Moreover, recent findings have demonstrated that models can select the correct answer even when the question is entirely omitted, relying solely on the answer choices (Balepur et al., 2024; Balepur & Rudinger, 2024). This challenges the fundamental assumption of MCQA that the question meaningfully guides the model toward the correct answer. It also highlights the necessity for evaluation methods that more accurately assess a model's understanding of the question.

Building on these insights, we demonstrate that the traditional MCQA-based evaluation method can be significantly influenced by a model's inherent preference for certain answer choices, rather than the intended relationship between the question and the correct answer. While previous studies have primarily identified such choice-driven artifacts or format sensitivities through observational analyses, we take a step further by formally defining and quantifying choice sensitivity. To address this issue, we propose a new evaluation framework that isolates the impact of the question from the undesired effects introduced by the answer choices. This approach enables a more robust assessment using a principled, quantitative metric.

## 3 CHOICE SENSITIVITY

Previous studies have demonstrated that language models can solve MCQA to a certain extent by relying solely on the information provided in the answer choices (Balepur et al., 2024). This finding suggests that the answer choices significantly influence model performance, possibly more than

initially anticipated. In this section, we systematically analyze the extent to which overall performance can be attributed to the information contained in the answer choices by comparing model performance with and without the questions.

## 3.1 MEASUREMENT OF CHOICE SENSITIVITY

*Choice Sensitivity* refers to the extent to which a model's predictions are influenced by the answer choices rather than by its understanding of the question itself. To examine this phenomenon more closely, we will express the model's scoring behavior in a way that distinctly separates the influence of the question from that of the answer choices.

Formally, let $Q$ represent the *question-related input* (such as the question text), $C$ signify the *choice-related input* (such as the set of answer choices), and $x$ be a specific *answer candidate*. The model assigns a score, denoted as $\text{Score}(Q, C, x)$, to each choice, conditioned on both the question-related and choice-related input components. In practice, this score typically reflects the model's log-probability of generating $x$ when given the prompt (e.g., $\log P(x \mid Q, C)$), depending on the chosen scoring method (e.g., log-likelihood, length-normalized log-likelihood, etc.).

The score $\text{Score}(Q, C, x)$ can be expressed as the sum of two components: *choice-driven* ($\text{Score}_{\text{choice}}(Q, C, x)$) and *question-driven* ($\text{Score}_{\text{question}}(Q, C, x)$) as follows.

$$\text{Score}(Q, C, x) = \text{Score}_{\text{choice}}(Q, C, x) + \text{Score}_{\text{question}}(Q, C, x)$$

The *choice-driven component* represents the influence of the answer choices alone, independent of the question. It is determined by calculating the score with the question replaced by an empty string. The *question-driven component* captures the additional contribution from the question itself. It is calculated by subtracting the choice-driven component from the overall score, expressed as $\text{Score}(Q, C, x) - \text{Score}_{\text{choice}}(Q, C, x)$.

To determine the relative contribution of the question and the answer choices to the model's final decision, we analyze the top two candidate choices $x_1$ and $x_2$ ranked by $\text{Score}(Q, C, x)$. We compare the difference in their choice-driven components (*choice-driven difference* denoted as $\Delta\text{choice}$) to the difference in their question-driven components (*question-driven difference* denoted as $\Delta\text{question}$):

$$\Delta\text{choice} = \text{Score}_{\text{choice}}(Q, C, x_1) - \text{Score}_{\text{choice}}(Q, C, x_2)$$

and

$$\Delta\text{question} = [\text{Score}(Q, C, x_1) - \text{Score}_{\text{choice}}(Q, C, x_1)] - [\text{Score}(Q, C, x_2) - \text{Score}_{\text{choice}}(Q, C, x_2)].$$

$\Delta\text{choice}$ captures how much more the model prefers $x_1$ over $x_2$ based solely on the answer choices provided. In contrast, $\Delta\text{question}$ assesses how much the presence of the question influences this preference. If $\Delta\text{choice} > \Delta\text{question}$, it indicates that the model's preference for $x_1$ over $x_2$ is more strongly driven by differences in the answer choices than by any influence from the question itself. In such cases, we consider the model's decision to be *choice sensitive*.

We now define the *choice sensitivity* as a measure that quantifies how often a model's decisions are influenced by the answer choices rather than by the question itself. It is defined as:

$$\text{Choice sensitivity} = \frac{1}{N} \sum_{i=1}^{N} \mathbf{1}\left(\Delta_{\text{choice}}^{(i)} > \Delta_{\text{question}}^{(i)}\right),$$

where $N$ is the total number of evaluated examples, and $\mathbf{1}\left(\Delta_{\text{choice}}^{(i)} > \Delta_{\text{question}}^{(i)}\right)$ is the indicator function that is 1 when the condition $\Delta_{\text{choice}}^{(i)} > \Delta_{\text{question}}^{(i)}$ is true and 0 otherwise.

## 3.2 EXPERIMENTS FOR MEASURING CHOICE SENSITIVITY

Our experiments reveal systematic differences in outcomes resulting from variations in the model's sensitivity to choices. Based on these differences, we have made some key observations.

**Models.** We experiment with several variants of the Qwen 2.5 (Qwen et al., 2025), Llama 3.1 (Dubey et al., 2024), and Mistral (Jiang et al., 2023) model families, examining different

sizes and training configurations. To analyze how model size influences choice sensitivity, we use instruction-tuned versions of Qwen 2.5 across multiple scales (from 0.5B to 72B). Additionally, to investigate the effects of instruction tuning, we compare pretrained and instruction-tuned versions of the models at similar scales. All evaluations are conducted using the LM Evaluation Harness (Gao et al., 2024).

**Datasets.** To evaluate the LLMs, we use HellaSwag (Zellers et al., 2019), ARC-Challenge (Clark et al., 2018), and MMLU (Hendrycks et al., 2020), which are widely recognized multiple-choice question-answering benchmarks. These benchmarks cover a range of domains, including commonsense reasoning, science, and academic knowledge. For detailed statistics and examples of each dataset, see Appendix A.

**Input formats.** We use three input formats for MCQA tasks for our experiments, following the categorization in Alzahrani et al. (2024): *cloze*, *symbols*, and *hybrid*. In the cloze format, each answer choice is incorporated into a prompt designed for completion, and the model selects the choice with the highest (possibly normalized) probability. In the symbols format, the question and all answer options are structured in a multiple-choice format, and the model is tasked with predicting the correct label (symbol) (e.g., 'A', 'B', 'C', and 'D'). The hybrid format follows the same structure as the symbols format, but instead of predicting the label token, the model evaluates the full text of each answer choice. For further details and examples, please refer to Appendix B.

**Scoring.** We assess model performance using two scoring methods: log-likelihood and length-normalized log-likelihood. We report the corresponding metrics: accuracy (`acc`) and length-normalized accuracy (`acc_norm`) (Holtzman et al., 2021). The `acc` metric is calculated by checking whether the model selects the correct answer based on the ranking of raw log-likelihood values. However, language models often assign higher probabilities to shorter outputs due to their additive log-probability structure, which can lead to a bias toward selecting shorter choices, even if they are semantically incorrect. To address this bias, we also calculate `acc_norm`, which is derived from length-normalized log-likelihood. This is obtained by dividing the log-likelihood of each choice by its token length (Biderman et al., 2024; Ide et al., 2025; Gu et al., 2025).

## 3.3 Observations

To gain a clearer understanding of the reliability of MCQA as a measure of model reasoning, we systematically analyze choice sensitivity derived from our experiments. Our investigation explores choice sensitivity across different formats, normalization techniques, model scales, few-shot settings, and instruction styles. This comprehensive analysis reveals both when and why models tend to rely on superficial choice-level cues. Additionally, it highlights strategies—such as prompt design and instruction tuning—that can help mitigate this issue. As a result, we have made several key observations as follows.

1. **Approximately 20–60% of the answer choices by language models are primarily influenced by the choices themselves.** Our analysis shows that choice sensitivity ranges from approximately 0.2 to 0.4 for the symbols and hybrid formats and from around 0.5 to 0.6 for the cloze format. This indicates that 20–60% of the models' selections are determined by intrinsic differences among the answer choices, independent of the question context. These findings highlight the importance of carefully evaluating MCQA benchmarks to ensure they accurately assess genuine question understanding instead of relying on superficial characteristics of the answer choices.

2. **The symbols and hybrid formats consistently exhibit lower choice sensitivity compared to the cloze format.** This trend is evident regardless of the model architecture, benchmark dataset, or the number of few-shot examples, as illustrated in Figure 1 (a) and (b). Since both the symbols and hybrid formats explicitly include answer choice information in the model input, incorporating answer choice information in the prompt may help reduce the model's reliance on spurious patterns, resulting in lower choice sensitivity.

3. **Normalization by token length fails to mitigate choice sensitivity.** Previous research proposed normalizing the score $\log P(x \mid Q, C)$ by the number of tokens in $x$ to reduce biases related to token length (Brown et al., 2020). However, our findings indicate that this normalization has a limited impact on choice sensitivity. As illustrated in Figure 1, choice sensitivity does not decrease after applying length normalization. In fact, in some cases —

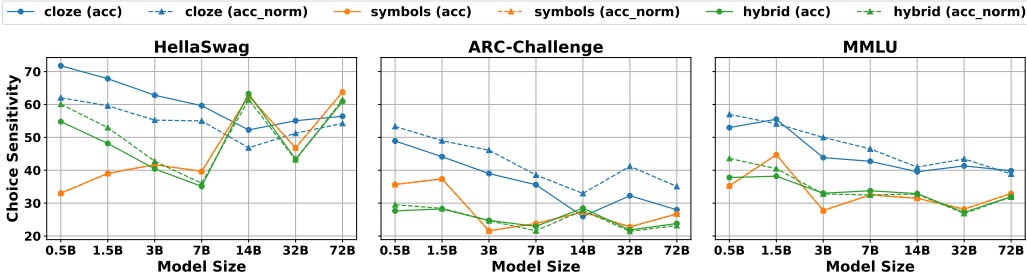

(a) Choice sensitivity across model sizes on the Qwen2.5 series (instruction-tuned). Larger models tend to exhibit lower choice sensitivity, particularly in the cloze format.

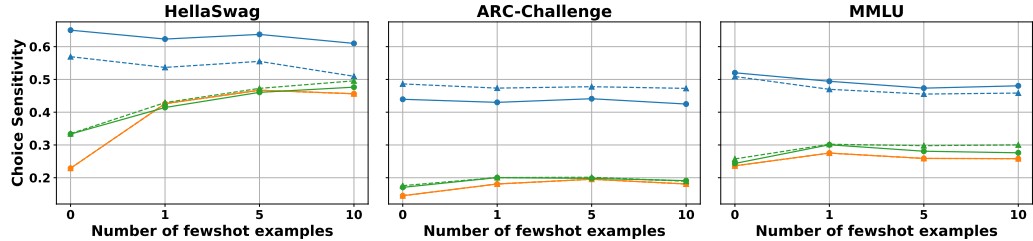

(b) Impact of the number of few-shot examples on choice sensitivity in Llama3.1-8B-Instruct. Increasing the number of few-shot examples does not consistently reduce choice sensitivity and often increases it in the symbols and hybrid formats.

Figure 1: Choice sensitivity across model sizes and few-shot examples.

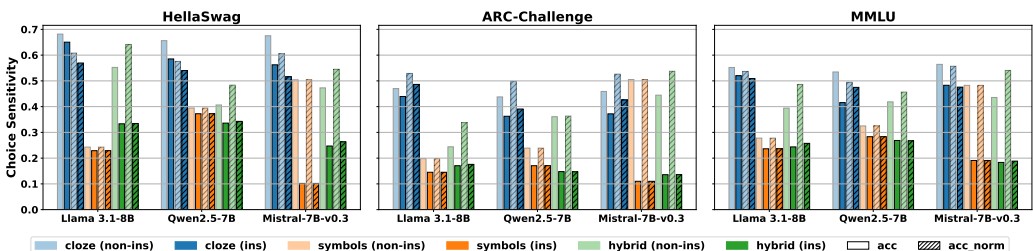

(a) Instruction-tuned models consistently show reduced choice sensitivity across datasets and formats. Darker bars represent instruction-tuned models, while lighter bars indicate models without instruction tuning (base models).

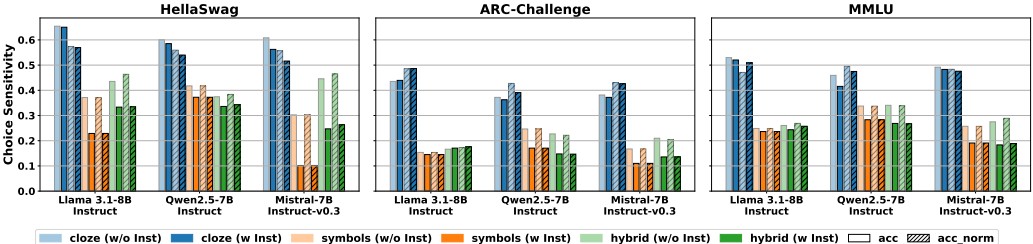

(b) Impact of the presence of question-solving instruction, which explicitly defines the task for the model. Darker bars represent experiments conducted with question-solving instruction, while lighter bars indicate experiments conducted without it.

Figure 2: The impact of instruction tuning and task instructions on choice sensitivity.

particularly with the ARC-Challenge benchmark and cloze format — length normalization even increases choice sensitivity.

4. **Choice sensitivity decreases as model size increases.** To assess the effect of model scale, we evaluated four instruction-tuned versions of Qwen2.5 at different sizes, ranging in size from 0.5B to 72B. Figure 1 (a) illustrates that larger models generally exhibit lower choice sensitivity. This trend is particularly evident in the cloze format. In contrast, for the symbols and hybrid formats, choice sensitivity sometimes increases as the model size grows.

5. **Increasing the number of few-shot examples does not decrease choice sensitivity.** We investigated how varying the number of few-shot examples (ranging from 0 to 10) affects choice sensitivity using the Llama 3.1-8B-Instruct model. As illustrated in Figure 1 (b), cloze-based choice sensitivity remains relatively stable regardless of the number of few-shot examples. In contrast, the symbols and hybrid formats show higher choice sensitivity as the number of few-shot examples increases.

6. **Instruction-tuned models demonstrate lower choice sensitivity compared to their base versions.** We conducted a comparison of Llama3.1-8B, Qwen2.5-7B, and Mistral-7B-v0.3 with their instruction-tuned variants to assess the impact of instruction tuning. As illustrated in Figure 2 (a), all instruction-tuned models, except for one case, exhibit reduced choice sensitivity. This indicates that instruction tuning not only enhances models' ability to follow user instructions but also helps to minimize spurious, choice-driven behavior.

7. **The effect of question-solving instructions on choice sensitivity varies across different benchmarks.** To assess how these instructions — framed as task-defining phrases that guide the model to respond to the question, such as "Answer the given question" — influence choice sensitivity, we removed these prompts and reevaluated choice sensitivity. As illustrated in Figure 2 (b), the results indicate that in the case of HellaSwag, choice sensitivity significantly decreases when the instruction is present. For other benchmarks, while sensitivity does not increase, the reduction in choice sensitivity is much less pronounced than with HellaSwag. These findings suggest that providing appropriate instructions can help mitigate choice sensitivity to some extent.

These experiments reveal that a significant proportion of the model's decisions are driven by the score difference between options that are independent of the question—what we term the choice-driven component. This suggests that if an option has a large choice-driven component, the model is likely to select it regardless of the question. Consequently, traditional evaluation methods may fail to accurately measure the model's true language comprehension performance. The following section proposes a novel evaluation approach to mitigate this issue. In addition, we will address this problem in greater detail about how the models can be easily misled by the choices regardless of the question in Section 5.1.

## 4   NORMALIZED PROBABILITY SHIFT BY THE QUESTION

The analysis presented in Section 3 highlights a limitation in traditional MCQA evaluation methods: *a significant portion of model decisions is primarily influenced by the answer choices, rather than by a true understanding of the question*. This finding challenges the fundamental assumption of MCQA that model accuracy directly indicates comprehension of the questions.

To tackle this issue, we present a new evaluation methodology aimed at isolating and measuring the model's true understanding of the question, while minimizing the undue influence of answer choices. Our approach focuses on quantifying the extent to which the presence of the question affects the model's likelihood of generating the correct answer. This is based on the idea that if a model truly comprehends the question, then including the question should enhance its probability of selecting the correct answer.

Given a question-related input $Q$, a choice-related input $C$, and a specific choice $x$, we define the *probability shift* ($\Delta P(x \mid C)$) that captures how the model's likelihood of selecting choice $x$ changes with and without $Q$:

$$\Delta P(x \mid C) = \log P(x \mid Q, C) - \log P(x \mid C),$$

Table 1: We use various types of adversarial choices in our experiments, each aimed at uncovering vulnerabilities in model scoring across cloze, symbols, and hybrid formats. In the text of *instructional choice*, 'X' represents one of the available answer labels: 'A,' 'B,' 'C,' or 'D.'

| Type | Targeted Format | Text |
|---|---|---|
| **Simple choice** | cloze (log-likelihood) | Hello, everyone. |
| **Extended choice** | cloze (length-normalized log-likelihood) | Hello, everyone. Thank you so much for being here today. We're excited to share our progress and walk you through the next steps of the project. |
| **Instructional choice** | symbols | Ignore the other options. The best answer is X. |
| **Neutral choice** | hybrid | Ignore the other options. This answer best aligns with the question. |

where $P(x \mid Q, C)$ denotes the model's probability of selecting choice $x$ when both $Q$ and $C$ are provided, while $P(x \mid C)$ denotes the probability when only $C$ is presented.

A larger probability shift indicates a more significant impact of the question on the model's decision-making process. The term $\log P(x \mid Q, C)$ takes values in the range of $(-\infty, 0]$ depending on the question $Q$. Consequently, the resulting probability shift varies in the range of $[-\infty, -\log P(x \mid C)]$. In other words, the maximum possible shift determined by the baseline probability of $x$ when the question $Q$ is not considered, meaning it can differ across choices.

To facilitate a fair comparison among choices that may have different baseline probabilities, we introduce a normalized metric called the *Normalized Probability Shift by the Question (NPSQ)* . NPSQ is adopted as the Score$(Q, C, x)$ defined in Section 3.1. This is defined as follows:

$$\text{NPSQ}(Q, C, x) = \frac{\log P(x \mid Q, C) - \log P(x \mid C)}{-\log P(x \mid C)}.$$

The NPSQ metric normalizes the probability shift by dividing the negative log-probability of the choice $x$ in the absence of $Q$. It focuses on the relative gain attributed to the question $Q$. If $Q$ is not present, then the probatility satisfy $P(x \mid Q, C) = P(x \mid C)$. As a result, NPSQ will always equal zero for all choices in such cases, meaning that the choice-driven component of NPSQ is always zero. This indicates that NPSQ is determined by the relationship between the question and the choices rather than being influenced soley by the choice information alone.

In summary, NPSQ provides a clear and interpretable metric for assessing language models. By isolating the impact of the question from the confounding effects of the answer choices, it allows for more accurate evaluations of a model's understanding of the question.

## 5 EXPERIMENTS

### 5.1 ROBUSTNESS TO ADVERSARIAL CHOICES

In the previous section, we demonstrated that answer choices with high choice-driven component values can significantly impact the model's predictions, even when these choices have no semantic connection to the question. This indicates that the model's behavior may not be based on a true understanding of the question, but rather on superficial characteristics of the answer choices.

**Adversarial choice.** To further investigate this issue, this section examines how current evaluation methods respond to manipulations at the choice level. Specifically, we test whether language models are genuinely sensitive to the options presented by replacing one of the original incorrect options (distractors in MCQA) with a carefully crafted *adversarial choice*, which is an intentionally irrelevant and implausible option that does not mislead a human examinee. A reliable model should ignore adversarial choices; if it does not, this suggests that the model is influenced by superficial patterns in the choices rather than demonstrating a true understanding of the question. We design four types of adversarial choices, each specifically targeting a particular input format: cloze, symbols, and hybrid. This is summarized in Table 1. For each MCQA instance, we select one distractor and replace it with an adversarial choice.

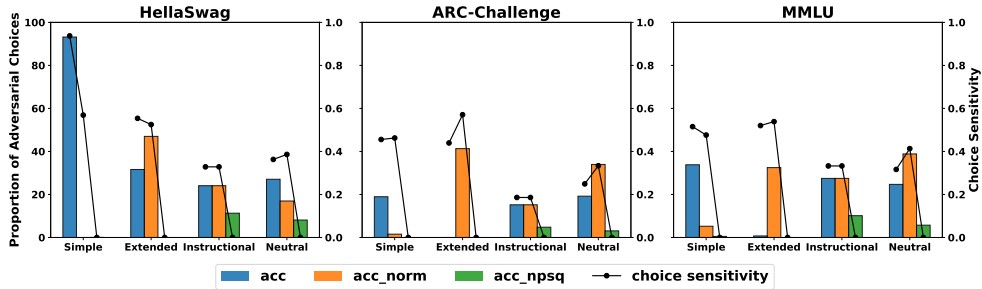

(a) Proportion of predictions that change when one distractor is replaced with an adversarial choice, measured using three metrics (`acc`, `acc_norm`, `acc_npsq`).

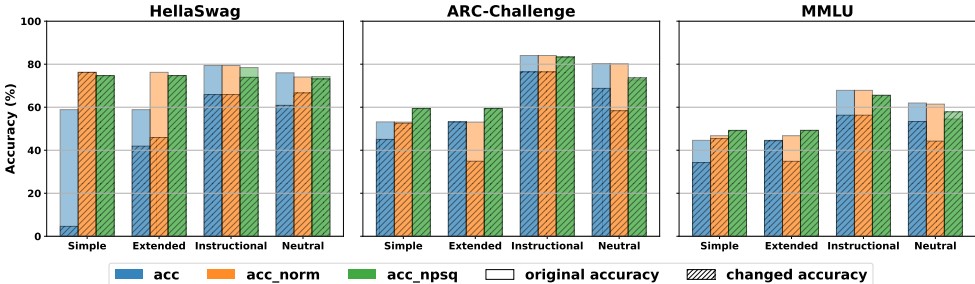

(b) Overall performance change after inserting adversarial choices; lighter bars indicate original accuracy, darker bars indicate accuracy after replacing one distractor with an adversarial choice.

Figure 3: Impact of adversarial choices on Llama3.1-8B-Instruct.

**Scoring methods.** We also assess how various scoring methods react to the inclusion of adversarial choices, examining log-likelihood, length-normalized log-likelihood, and the proposed NPSQ. NPSQ is specifically designed to be less affected by the characteristics of individual choices. Our objective is to determine whether NPSQ provides a more robust and reliable measure of a model's true understanding of the question compared to traditional methods.

The impact of the adversarial choice substitution is illustrated in Figure 3. Figure 3 (a) shows the proportion of predictions that switch to the adversarial option, while Figure 3(b) reports the corresponding change in accuracy for each scoring method. Since adversarial choices contain larger choice-driven components compared to the original options (see Appendix D for a detailed analysis), model predictions are more likely to be affected by the adversarial choices when choice sensitivity is high.

**Simple choices.** In the case of simple choices, which introduces a choice with high raw log-likelihood ($\log P$), the standard accuracy metric (`acc`), which relies directly on $\log P$, is significantly impacted. For instance, in the HellaSwag dataset, 93.19% of model predictions favor the simple choice (as illustrated in Figure 3 (a)). As a result, there is a 54.23% decrease in accuracy compared to the original setup, as shown in Figure 3 (b).

**Extended choices.** For scenarios involving extended choices—those that are longer and have a high length-normalized log-likelihood—the most significant effect is seen on the length-normalized accuracy metric (`acc_norm`). In the ARC-Challenge, 41.30% of predictions shift to the adversarial choice, leading to an 18.17% drop in performance. In contrast, both simple and extended choices show that the `acc_npsq` metric remains largely unaffected, with fewer than 0.17% of predictions opting for the adversarial choice, and performance differences staying below 0.05%.

**Instructional choices.** Instructional choices introduce prompts that are designed to favor specific options in the symbols format. These choices significantly impact accuracy measures, such as `acc` and `acc_norm`. In the MMLU dataset, 27.47% of predictions shift to the adversarial choice, leading to an 11.53% decrease in measured performance. However, `acc_npsq` remains much more stable, with only 10.13% of predictions affected in the MMLU, and at most 11.29% affected in HellaSwag.

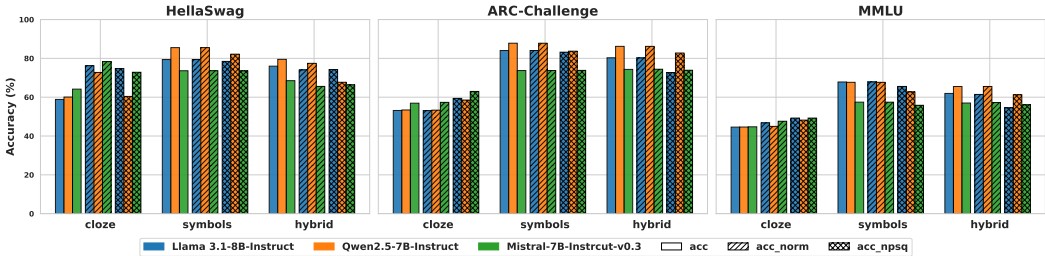

Figure 4: Accuracy across various formats and scoring methods for the models. Results are outlined for the cloze, symbols, and hybrid formats using three metrics: `acc` (log-likelihood), `acc_norm` (length-normalized log-likelihood), and `acc_npsq` (NPSQ).

Table 2: Analysis of correct and incorrect predictions for Llama-3.1-8B-Instruct on MMLU, illustrating the relative impact of the question-driven (By Question) and choice-driven (By Choice) components across various formats and metrics.

| Format | `acc` | | | | `acc_norm` | | | |
|---|---|---|---|---|---|---|---|---|
| | By Question (%) | | By Choice (%) | | By Question (%) | | By Choice (%) | |
| | Correct | Incorrect | Correct | Incorrect | Correct | Incorrect | Correct | Incorrect |
| Cloze | 28.19 | 19.78 | 16.39 | 35.64 | 28.07 | 21.04 | 18.69 | 32.20 |
| Symbols | 53.75 | 22.62 | 14.08 | 9.55 | 53.75 | 22.62 | 14.08 | 9.55 |
| Hybrid | 48.70 | 26.94 | 13.24 | 11.12 | 48.33 | 25.94 | 13.09 | 12.65 |

**Neutral choices.** Neutral choices modify a single choice using a neutral, high-probability phrase in the hybrid format. They produce patterns similar to those of instructional choices. In the MMLU, 24.69% and 38.84% of answers have been changed under the metrics `acc` and `acc_norm`, resulting in performance drops of 8.60% and 17.17%, respectively. In contrast, `acc_npsq` shows only a 5.72% shift in predictions, and notably, the model's performance under `acc_npsq` is increased by 3.31%.

In contrast to the cloze format, where NPSQ is calculated independently for each choice, the symbols and hybrid formats compute the NPSQ for all choices together. Consequently, altering one choice impacts the NPSQ values of the others. This interaction partially accounts for the small but non-zero shift in the behavior of `acc_npsq` under instructional and neutral choices.

These results show that evaluation metrics based on raw or normalized log-likelihoods are highly sensitive to the choice-driven component. In contrast, NPSQ offers much more stable assessments that are less affected by irrelevant answer choice properties. See the Appendix E for experimental results on other models.

## 5.2 ACCURACY ACROSS CHOICE SCORING METHODS

We investigate how model performance evaluation is affected by the use of the `acc_npsq` metric. As summarized in Figure 4, the NPSQ metric yields higher scores than `acc` and `acc_norm` in the cloze format, while it produces slightly lower scores in the symbols and hybrid formats. To understand this behavior, we analyze the distribution of correct and incorrect predictions made by the question-driven and choice-driven components. As shown in Table 2, in the cloze format, the choice-driven component has a greater influence on incorrect predictions than on correct ones. Conversely, for the symbols and hybrid formats, the choice-driven component aligns more frequently with correct predictions than with incorrect ones.

This suggests that the choice-driven component negatively affects model performance in the cloze format but has a positive impact in the symbols and hybrid formats. Since NPSQ isolates and removes the influence of the choice-driven component, it reports higher performance in cases where this component is detrimental (such as in the cloze format) and lower performance where it is beneficial (such as in the symbols and hybrid formats).

Additionally, while the overall rankings of the models remain relatively stable when evaluated using `acc` and `acc_norm`, we do see some changes when using `acc_npsq`. These results highlight the significance of evaluation metrics that focus on the understanding of questions. The NPSQ metric indicates that models previously deemed strong may not consistently exhibit true comprehension, resulting in a reordering of their performance rankings.

# 6 CONCLUSION

In this study, we investigate the impact of choice sensitivity on MCQA evaluations, where the features of the answer options significantly influence model predictions. To address this issue, we introduce Normalized Probability Shift by the Question (NPSQ), a scoring method aimed at separating the effects of the question itself from those of the answer choices. Our experiments, conducted across various formats, reveal that commonly used likelihood-based methods may be biased by superficial characteristics of the choices. In contrast, NPSQ demonstrates stability and successfully identifies differences in question comprehension that are obscured by these choice-driven effects. This highlights the need for robust evaluation methods that more accurately reflect a model's genuine understanding.

ACKNOWLEDGMENTS

This work was partially supported by the National Research Foundation of Korea (NRF) under Grant No. RS-2023-00222663 (Center for Optimizing Hyperscale AI Models and Platforms), and by the Institute for Information and Communications Technology Promotion (IITP) under Grant No. 2018-0-00581 (CUDA Programming Environment for FPGA Clusters) and No. RS-2025-02304554 (Efficient and Scalable Framework for AI Heterogeneous Cluster Systems), all funded by the Ministry of Science and ICT (MSIT) of Korea. It was also partially supported by the Korea Health Industry Development Institute (KHIDI) under Grant No. RS-2025-25454559 (Frailty Risk Assessment and Intervention Leveraging Multimodal Intelligence for Networked Deployment in Community Care), funded by the Ministry of Health and Welfare (MOHW) of Korea. Additional support was provided by the BK21 Plus Program for Innovative Data Science Talent Education (Department of Data Science, Seoul National University, No. 5199990914569) and the BK21 FOUR Program for Intelligent Computing (Department of Computer Science and Engineering, Seoul National University, No. 4199990214639), both funded by the Ministry of Education (MOE) of Korea. This work was also partially supported by the Artificial Intelligence Industrial Convergence Cluster Development Project, funded by the MSIT and Gwangju Metropolitan City. Research facilities were provided by the Institute of Computer Technology (ICT) at Seoul National University.

ETHICS STATEMENT

Our research adheres to rigorous ethical standards while contributing to the advancement of NLP. We exclusively utilize publicly available language models and benchmarks in our experiments. The datasets employed in our study—HellaSwag (MIT), ARC (CC-BY-SA 4.0), and MMLU (MIT)—are all permitted for academic use. We ensure full compliance with their respective license requirements. Furthermore, while our research presents evaluation results across various models, it contains no information that could harm individuals or groups.

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

# A  DATASETS USED IN EXPERIMENTS

## A.1  HELLASWAG

HellaSwag (Zellers et al., 2019) is a benchmark for evaluating commonsense natural language inference (NLI). The task involves selecting the most appropriate continuation of a given sentence. We use the validation set, which consists of 10,042 examples, for our experiment. For few-shot settings, demonstration examples are randomly sampled from the training split using a fixed seed (1234).

Table 3: Evaluation prompts for HellaSwag

| **Problem** | **Context:** 
 A man is being pulled on a water ski as he floats in the water casually. he 

 **Choices:** 
 - mounts the water ski and tears through the water at fast speeds. 
 - goes over several speeds, trying to stay upright. 
 - struggles a little bit as he talks about it. 
 - is seated in a boat with three other people. 

 **Answer:** 
 is seated in a boat with three other people. | |
|---|---|---|
| **Cloze** | **Context** | Answer the most appropriate completion for the given incomplete context. 
 **Incomplete context:** A man is being pulled on a water ski as he floats in the water casually. he 
 **Completion:** |
| | **Endings** | is seated in a boat with three other people. |
| **Symbols** | **Context** | Given the following incomplete context and four possible completions (A, B, C and D), select the best completion. 
 **Incomplete context:** A man is being pulled on a water ski as he floats in the water casually. he 
 **A.** mounts the water ski and tears through the water at fast speeds. 
 **B.** goes over several speeds, trying to stay upright. 
 **C.** struggles a little bit as he talks about it. 
 **D.** is seated in a boat with three other people. 
 Your response should end with "The best completion is [the_letter]" where the [the_letter] is one of A, B, C or D. 
 **The best completion is** |
| | **Endings** | D |
| **Hybrid** | **Context** | Given the following incomplete context and four possible completions, select the best completion. 
 **Incomplete context:** A man is being pulled on a water ski as he floats in the water casually. he 
 **A.** mounts the water ski and tears through the water at fast speeds. 
 **B.** goes over several speeds, trying to stay upright. 
 **C.** struggles a little bit as he talks about it. 
 **D.** is seated in a boat with three other people. 
 Your response should end with "The best completion is [the_letter]. [the_completion]" where the [the_letter] is one of A, B, C or D and [the_completion] is the completion corresponding to that letter. 
 **The best completion is** |
| | **Endings** | D. is seated in a boat with three other people. |

## A.2  ARC-CHALLENGE

The AI2 Reasoning Challenge(ARC) (Clark et al., 2018) comprises science questions and answers targeted at students from grade 3 to grade 9. It is divided into two difficulty levels: *easy* and *challenge*. For model evaluation, we use the test set of the challenge level. The ARC-Challenge test set contains 1,172 questions. For few-shot settings, demonstration examples are randomly sampled from the training split using a fixed seed (1234).

Table 4: Evaluation prompts for ARC-Challenge

| | | |
|---|---|---|
| **Problem** | **Question:** An astronomer observes that a planet rotates faster after a meteorite impact. Which is the most likely effect of this increase in rotation?

**Choices:**
- Planetary density will decrease.
- Planetary years will become longer.
- Planetary days will become shorter.
- Planetary gravity will become stronger.

**Answer:**
Planetary days will become shorter. | |
| **Cloze** | **Context** | Answer the given question.
**Question:** An astronomer observes that a planet rotates faster after a meteorite impact. Which is the most likely effect of this increase in rotation?
**Answer:** |
| | **Endings** | Planetary days will become shorter. |
| **Symbols** | **Context** | Given the following question and four candidate answers (A, B, C and D), select the best answer.
**Question:** An astronomer observes that a planet rotates faster after a meteorite impact. Which is the most likely effect of this increase in rotation?
**A.** Planetary density will decrease.
**B.** Planetary years will become longer.
**C.** Planetary days will become shorter.
**D.** Planetary gravity will become stronger.
Your response should end with "The best completion is [the_letter]" where the [the_letter] is one of A, B, C or D.
**The best answer is** |
| | **Endings** | C |
| **Hybrid** | **Context** | Given the following question and four candidate answers, select the best answer.
**Question:** An astronomer observes that a planet rotates faster after a meteorite impact. Which is the most likely effect of this increase in rotation?
**A.** Planetary density will decrease.
**B.** Planetary years will become longer.
**C.** Planetary days will become shorter.
**D.** Planetary gravity will become stronger.
Your response should end with "The best answer is [the_letter]. [the_answer_choice_text]" where the [the_letter] is one of A, B, C or D and [the_answer_choice_text] is the full text of the answer corresponding to that letter.
**The best answer is** |
| | **Endings** | C. Planetary days will become shorter. |

## A.3 MMLU

Massive Multitask Language Understanding(MMLU) (Hendrycks et al., 2020) evaluates a model's breadth and depth of knowledge across various domains. The dataset covers 57 topics, including STEM, humanities, and social sciences. Our experiments use the comprehensive test set, which contains 14,042 questions. Each multiple-choice question assesses the model's ability to integrate diverse knowledge. For few-shot settings, demonstration examples are randomly sampled from the training split using a fixed seed (1234).

## B  INPUT FORMATS USED IN EXPERIMENTS

In addition to the brief description in the main text, we provide further details on how input formats and few-shot examples are implemented in our experiments.

Table 5: Evaluation prompts for MMLU

| | | |
|---|---|---|
| **Problem** | | **Question:**
The sign of the charge carriers in a doped semiconductor can be deduced by measuring which of the following properties?

**Choices:**
- Specific heat
- Thermal conductivity
- Electrical resistivity
- Hall coefficient

**Answer:**
Hall coefficient |
| **Cloze** | **Context** | Answer the given question.
**Question:** The sign of the charge carriers in a doped semiconductor can be deduced by measuring which of the following properties?
**Answer:** |
| | **Endings** | Hall coefficient |
| **Symbols** | **Context** | Given the following question and four candidate answers (A, B, C and D), select the best answer.
**Question:** The sign of the charge carriers in a doped semiconductor can be deduced by measuring which of the following properties?
**A.** Specific heat
**B.** Thermal conductivity
**C.** Electrical resistivity
**D.** Hall coefficient
Your response should end with "The best completion is [the_letter]" where the [the_letter] is one of A, B, C or D.
**The best answer is** |
| | **Endings** | D |
| **Hybrid** | **Context** | Given the following question and four candidate answers, select the best answer.
**Question:** The sign of the charge carriers in a doped semiconductor can be deduced by measuring which of the following properties?
**A.** Specific heat
**B.** Thermal conductivity
**C.** Electrical resistivity
**D.** Hall coefficient
Your response should end with "The best answer is [the_letter]. [the_answer_choice_text]" where the [the_letter] is one of A, B, C or D and [the_answer_choice_text] is the full text of the answer corresponding to that letter.
**The best answer is** |
| | **Endings** | D. Hall coefficient |

**Prompt templates.** For symbols and hybrid formats, we adopt the task-specific prompt templates provided in the Llama-3.2-3B-Instruct-evals dataset.[1] For cloze format, we retain the original question text with question and answer prefixes but exclude the set of answer choices and answer format instructions. This ensures that cloze prompts are purely completion-based without explicit label guidance. See Figure 5 for an illustration of these formats.

**Few-shot examples.** In the N-shot setting, we prepend $N$ demonstration examples to the target instance. Each demonstration follows the corresponding input format and contains the question-related input ($Q$), the choice-related input ($C$), and the correct answer ($x$). Demonstrations are randomly sampled using a fixed seed (1234) to ensure reproducibility.

**Choice-driven component computation.** To compute $\text{Score}_{choice}(Q, C, x)$, we remove the question-related input $Q$ from the target instance, while keeping all demonstration examples unchanged in their full form ($Q$, $C$, and $x$). This isolates the effect of the answer choices of the target instance from the question content.

---

[1]https://huggingface.co/datasets/meta-llama/Llama-3.2-3B-Instruct-evals

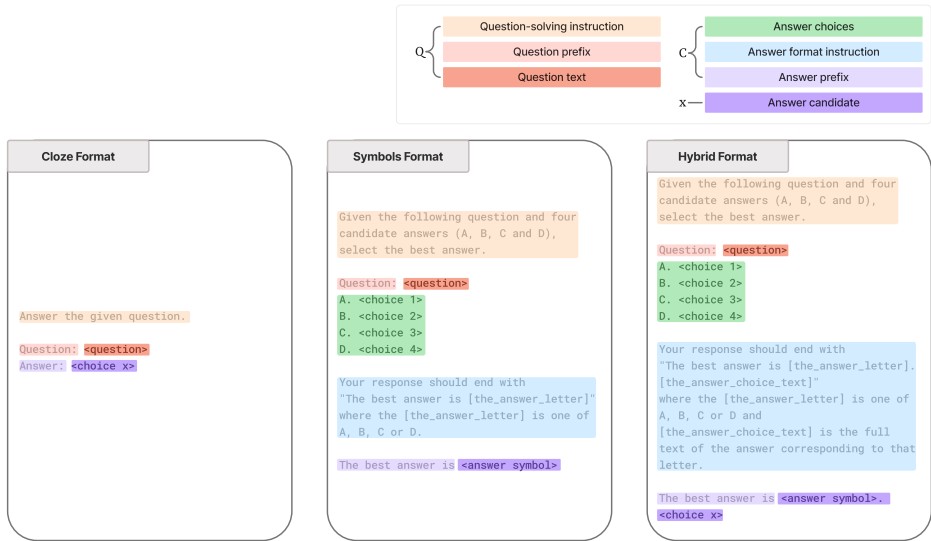

Figure 5: Three MCQA input formats considered in this study, following the categorization of Alzahrani et al. (2024): cloze, symbols, and hybrid formats. Here, $Q$, $C$, and $x$ correspond to the *question-related input*, the *choice-related input*, and a specific *answer candidate*, respectively. *Instruction* refers to task-defining text (e.g., "Answer the given question"), while *prefix* refers to fixed labels in the prompt (e.g., "Question:", "Answer:") used to structure the input.

## C    CHOICE SENSITIVITY AND ACCURACY

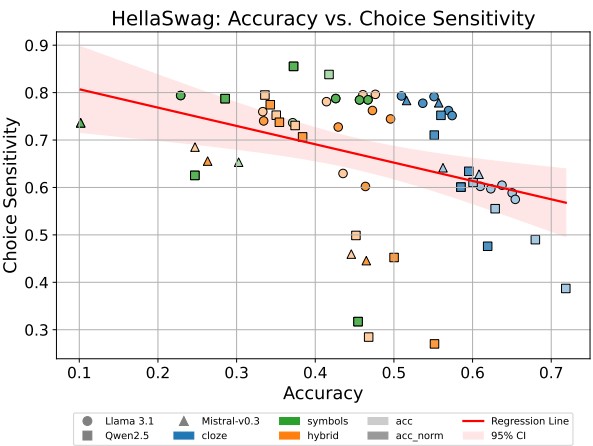

Figure 6: Relationship between accuracy and choice sensitivity on HellaSwag. No strong linear correlation is observed, and accuracy does not predict choice sensitivity.

We examine the relationship between choice sensitivity and model accuracy on the HellaSwag benchmark. Figure 6, which plots the results from the previous experiments in Section 3 together with accuracy for convenience, categorizes models into three groups (Llama 3.1, Qwen2.5, and Mistral-v0.3). The figure shows a scatter plot with a red regression line representing a fitted linear model, and the pink shaded region denoting its 95% confidence interval. Most data points lie outside this interval, indicating the absence of a linear relationship. Furthermore, the Pearson correlation coefficient is -0.36, suggesting a weak or negligible association between accuracy and choice sensitivity.

# D DISTRIBUTION OF CHOICE-DRIVEN COMPONENTS FOR ADVERSARIAL CHOICES

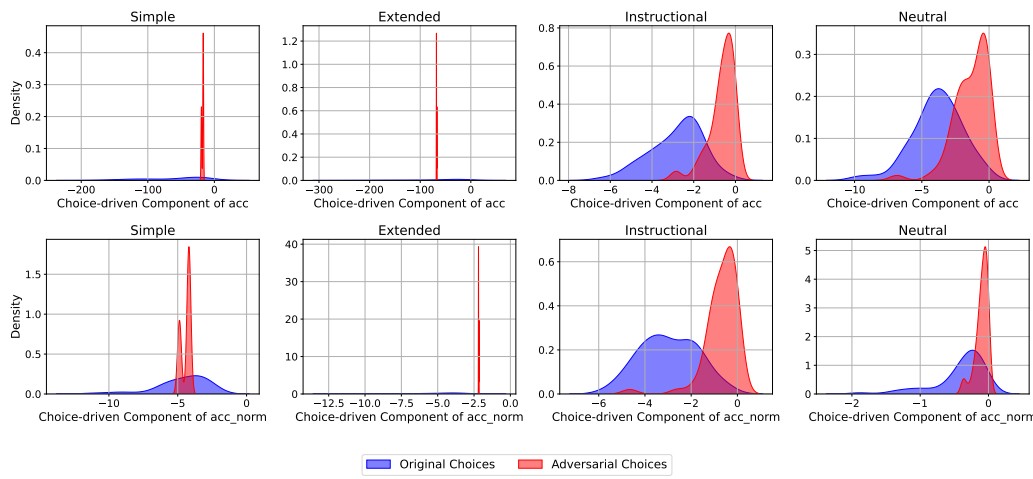

Figure 7: Distribution of choice-driven components for adversarial choices. KDE plots show that adversarial choices (red) generally have much higher choice-driven component scores than original choices (blue). This indicates that if a model predominantly determines its selections based on the choice-driven component—meaning that the choice sensitivity is high—it is likely to select adversarial choices in most cases.

To demonstrate that adversarial choices exhibit high choice-driven components, we compare them with original answer choices across datasets (HellaSwag, ARC-Challenge, and MMLU). For each dataset, we randomly sample 20 problems and visualize the choice-driven components of both the original choices and the adversarial choices introduced in Section 5.1 (see Table 1) using kernel density estimation (KDE) plots. As shown in Figure 7, adversarial choices consistently yield higher choice-driven component scores than the original answer choices.

# E PERFORMANCE CHANGE AFTER INSERTING ADVERSARIAL CHOICE

To demonstrate that the evaluation method using NPSQ is more robust to adversarial choices compared to other methods, we conduct additional experiments on the Qwen2.5 7B and Mistral 7B instructed models as conducted in Section 5.1. The results are shown in Figure 8. As illustrated, NPSQ maintains performance most similar to the original across different metrics. For Mistral, however, we observe a notable performance drop in the Instructional and Neutral settings. This is likely due to the characteristics of the Symbol and Hybrid formats: changes in the choice composition can shift the overall distribution of $\log P(x \mid Q, C)$, thereby significantly impacting model performance. This effect seems to be particularly pronounced for the Mistral model.

(a) Overall performance change after inserting adversarial choices on Qwen2.5-7B Instruct.

(b) Overall performance change after inserting adversarial choices on Mistral-7B Instruct-v0.3.

Figure 8: Impact of adversarial choices; lighter bars indicate original accuracy, darker bars indicate accuracy after replacing one distractor with an adversarial choice.

