# OpenReview forum: "Choices Speak Louder than Questions"
_ICLR.cc/2026/Conference — ICLR 2026 Poster_

### Official Review · Reviewer_WE2p · 2025-10-28

**Soundness:** 2
**Presentation:** 3
**Contribution:** 2
**Rating:** 6
**Confidence:** 4

**Summary:**

In this paper, authors raise the issue of LLM answers to questions with option choice being driven by some intrinsic hints in the option texts, not by understanding of the question. Authors decompose the log-probability that model assign to each answer to the given question into choice- and question-driven components and develop a method to identify questions for which the model's answers are caused by choice texts. Authors apply this method to analyze several LLMs (base and instruction-tuned) on three common MCQA benchmarks. Finally, they propose a new method to select the answer option (NPSQ) that is less sensitive to superficial characteristics of the answer choices than log-probability and provide analysis to show that it is a more reliable tool to assess the model's comprehension of the question.

**Strengths:**

- This paper addresses an important problem; it proposes a novel approach to identify and mitigate such "choice-induced" biases; this method is easy to deploy and does not require training of a model or additional data.

- The proposed method can be used to improve existing (and future) MCQA benchmarks by removing "shortcuts" presented in the formulation of choice option texts.

- Experiments cover different task formulations (options only/cloze prompting/hybrid). A detailed analysis of the proposed method (NPSQ) and observed effects is provided.

- Paper is well-written and easy to follow.

**Weaknesses:**

- My main concern regarding the proposed method is the fact that it does not take into account effects caused by the order of the options (which is known from the previous studies in the field to be an influential factor). In several works it was shown that, given a list, LLMs often `focuses' more on the later entries from it than on earlier. It can (theoretically) affect the proposed sensitivity analysis method:

Choice sensitivity inequality for one question (line 140) can be reformulated into
$$
2 * (Score_{choice}(Q, C, x_1) - Score_{choice}(Q, C, x_2) ) > Score(Q, C, x_1) - Score(Q, C, x_2)
$$
if $x_1$ is the last option (e.g., D for 4-option setup in MMLU), and $x_2$ is one of the first, left side may be very high, because without question (empty "C" in calculation of $Score_{choice}$) mentioned above effect of list item recall may take place.

A similar issue arises in the formula for NPSQ (line 321).

 I would recommend to further analyze this issue and, maybe, (although it is just one of the potential ways to address it, if it is a real problem; I do not insist on using this suggestion) average probabilities from initial options order and after a certain permutation.

- The performed experiments cover only relatively small models up to 8B parameters. Presented results show that larger model has smaller (but still noticeable) sensitivity bias, and it would be very interesting to see the results for a larger model (at least -14B or -32B).

**Questions:**

- Would inclusion of irrelevant (i.e., ) "uncertainty sinks" options (like ``I don't know`` or ``None of the above``) affect the model's sensitivity to options?

---

> ### Author Response · Authors · 2025-11-21
>
> We appreciate your thoughtful feedback. We address the individual comments below:
>
> Q1: The proposed method does not consider permutation bias in LLMs, which refers to the problem where model performance varies significantly based on the choice order
>
> A1: While we acknowledge the presence of permutation bias, we chose not to address it in our method because we believe it is not primarily driven by order-dependent variations in the choice-driven component—the aspect our method focuses on.
>
> To support this claim, we conducted the following experiment. Assume that the primary reason model performance changes when the option order is shuffled is that reordering substantially alters the choice-driven components of the options. Under this assumption, if moving the correct answer from position A to position D increases performance, this improvement should stem from an increased choice-driven component for the correct answer, which should in turn lead to higher choice sensitivity.
>
> We therefore measured both accuracy and choice sensitivity when the correct answer appeared in position A versus position D. However, our results show that choice sensitivity does not consistently increase when performance increases. This suggests that order-driven changes in the choice-driven component are not a major factor in performance increase.
>
> | **Benchmark**     | **Eval Format** | **Answer Position** | **Choice Sensitivity** | **Performance** |
> | ----------------- | --------------- | ------------------- | ---------------------- | --------------- |
> | **Hellaswag**     | symbol          | A                   | 26.95                  | 75.12           |
> |                   |                 | D                   | 20.25                  | 82.57           |
> |                   | hybrid          | A                   | 30.62                  | 70.32           |
> |                   |                 | D                   | 38.49                  | 80.67           |
> | **ARC-Challenge** | symbol          | A                   | 16.55                  | 82.94           |
> |                   |                 | D                   | 12.54                  | 82.85           |
> |                   | hybrid          | A                   | 17.83                  | 84.73           |
> |                   |                 | D                   | 17.83                  | 72.10           |
> | **MMLU**          | symbol          | A                   | 25.50                  | 66.02           |
> |                   |                 | D                   | 21.77                  | 64.82           |
> |                   | hybrid          | A                   | 25.36                  | 69.98           |
> |                   |                 | D                   | 25.75                  | 48.06           |
>
>
>
> Q2: The experiments are limited to models up to 8B parameters, and evaluating larger models (14B–32B) could provide additional insights.
>
> A2: We have added results for Qwen2.5 14B, 32B and 72B. Please refer to Figure 1(a) in the revised version.
>
> Q3: How do irrelevant “uncertainty sinks” (e.g., I don’t know, None of the above) affect the model’s sensitivity to options?
>
> A3: We tested this by adding “I don’t know” as an uncertainty-sink option. The results show that this option has minimal impact on both choice sensitivity and model accuracy. This is likely because its choice-driven component is relatively small or comparable to other options.
>
> | **Benchmark**     | **Eval Format** | **Uncertainty Sinks** | **Choice Sensitivity** | **Performance** |
> | ----------------- | --------------- | --------------------- | ---------------------- | --------------- |
> | **Hellaswag**     | symbol          | w                     | 22.89                  | 79.39           |
> |                   |                 | w/o                   | 27.66                  | 82.89           |
> |                   | hybrid          | w                     | 33.33                  | 75.98           |
> |                   |                 | w/o                   | 31.74                  | 79.91           |
> | **ARC-Challenge** | symbol          | w                     | 14.51                  | 84.04           |
> |                   |                 | w/o                   | 16.13                  | 85.75           |
> |                   | hybrid          | w                     | 17.06                  | 80.29           |
> |                   |                 | w/o                   | 19.28                  | 76.11           |
> | **MMLU**          | symbol          | w                     | 23.63                  | 67.83           |
> |                   |                 | w/o                   | 23.52                  | 71.41           |
> |                   | hybrid          | w                     | 24.36                  | 61.94           |
> |                   |                 | w/o                   | 25.25                  | 56.79           |

---

### Official Review · Reviewer_yhhk · 2025-10-31

**Soundness:** 3
**Presentation:** 3
**Contribution:** 3
**Rating:** 6
**Confidence:** 3

**Summary:**

This paper investigates "choice sensitivity" in the MCQA evaluation of LLMs: the phenomenon where models exploit superficial features of the answer options rather than genuinely comprehending the question. The authors demonstrate that a significant portion of model decisions (20-60%) is driven by this sensitivity. To address this, they propose a new evaluation metric, Normalized Probability Shift by the Question (NPSQ), which is designed to isolate the impact of the question itself. Experiments, particularly those using adversarial choices, show that NPSQ is significantly more robust to option-based artifacts than traditional log-likelihood or length-normalized metrics.

**Strengths:**

The paper addresses a clear and increasingly important issue in LLM evaluation. As models achieve high scores on benchmarks, it is crucial to understand if this reflects true comprehension or artifact exploitation.

The proposed NPSQ metric is intuitive, well-motivated, and directly targets the identified problem by quantifying the "value" of the question.

The use of adversarial choices provides a very clear and convincing demonstration of the weaknesses of existing metrics and the robustness of NPSQ. The results in Figure 3 are particularly compelling.

**Weaknesses:**

While valuable, the contribution is an incremental improvement in evaluation methodology rather than a new task, model, and with no fundamental insight into model reasoning.

The analysis of why models exhibit this sensitivity is not deeply explored, though this is not the primary focus. The experiments are solid but could be extended to a wider range of model architectures and benchmarks.

**Questions:**

N.A.

---

> ### Author Response · Authors · 2025-11-21
>
> Thank you for the thoughtful comment. Identifying the fundamental cause of choice sensitivity is indeed important, but it is not the central focus of this paper. We acknowledge the value of this direction and plan to explore it in future work.

---

### Official Review · Reviewer_ACim · 2025-10-31

**Soundness:** 3
**Presentation:** 3
**Contribution:** 3
**Rating:** 6
**Confidence:** 3

**Summary:**

This paper argues that standard MCQA metrics reward models for exploiting "option-surface cues" rather than understanding the question. It proposes NPSQ (Normalized Probability Shift by the Question), a new metric that isolates the question's contribution by comparing an option's log-likelihood with versus without the question, then normalizing this gain. Experiments on HellaSwag, ARC-Challenge, and MMLU show that while standard accuracy is fragile to adversarial or rephrased choices, NPSQ accuracy remains stable and can reorder model rankings. The study also quantifies this "choice sensitivity" and explores other factors like formats and prompts.

**Strengths:**

1. The paper identifies and formalizes a pervasive evaluation artifact, choice sensitivity, and gives a principled, testable metric to mitigate it.

2. The core construct (question-conditioned vs. question-ablated likelihood shift with normalization) is simple, auditable, and easy to slot into existing LM-eval pipelines.

**Weaknesses:**

1. The normalization in NPSQ is not stress-tested against plausible alternatives (e.g., z-scores, temperature scaling, ECE), leaving ranking stability under-substantiated.

2. Key stability claims (flip rates, adversarial drops) lack uncertainty quantification and significance testing, weakening the statistical support for the conclusions.

3. The metric relies on token-level probabilities and a hand-crafted “no-question” template whose wording or API backend may change outcomes, reducing reproducibility.

4. Computing scores with and without the question doubles evaluation cost, which is non-trivial for large suites.

**Questions:**

See Weakness

---

> ### Author Response · Authors · 2025-11-21
>
> Thank you for taking the time to review our paper. We address the individual comments below:
>
> Q1: The normalization in NPSQ is not stress-tested against plausible alternatives (e.g., z-scores, temperature scaling, ECE), leaving ranking stability under-substantiated.
>
> A1: NPSQ is designed to compare changes in log-probability caused specifically by the presence or absence of the question. For this purpose, min–max normalization is appropriate method because it aligns the scales of the maximum and minimum changes. (Since the minimum log-probability change is −∞, our implementation normalizes based on the maximum range only.)
> While other normalization schemes exist, such as z-scores or temperature scaling, they are not intended for ensuring fair comparability of magnitude shifts in log-probability. Therefore, we believe min–max normalization is the most suitable choice in this context.
>
> Q2: Key stability claims (flip rates, adversarial drops) lack uncertainty quantification and significance testing, weakening the statistical support for the conclusions.
>
> A2: In our settings (fixed input template, zero-shot setting), language model evaluations become fully deterministic, producing identical outputs across repeated runs. Because there is no stochastic variation in the predictions, classical uncertainty quantification methods—such as confidence intervals, t-tests, or other significance tests—cannot be meaningfully applied. These techniques rely on sampling variability, which is absent in this evaluation setting.
> Instead, to ensure that our observations are not artifacts of a particular setup, we conducted experiments across a diverse set of models, benchmarks, and evaluation formats. This demonstrates that the reported stability phenomena are not restricted to a narrow set of conditions.
> We acknowledge, however, that Section 5.1 was originally conducted only with Llama-3.1. To address this limitation, we performed additional experiments using Qwen2.5 and Mistral models. The results have been added to the Appendix as supplementary evidence.
>
> Q3: The metric relies on token-level probabilities and a hand-crafted “no-question” template whose wording or API backend may change outcomes, reducing reproducibility.
>
> A3: In MCQA benchmarks, the question component is clearly defined. As long as the evaluation template follows the benchmark’s definition, removing the question yields a natural and unambiguous no-question template. Therefore, we believe reproducibility concerns are limited.
>
> Q4: Computing scores with and without the question doubles evaluation cost, which is non-trivial for large suites.
>
> A4: Current MCQA benchmarks already require increased evaluation cost to address other issues such as permutation bias[1,2]—often requiring more than 2× computation. In comparison, our method only needs a 2× cost increase, which we believe is a reasonable overhead.
>
>
> > [1] Zheng, Chujie, et al. "Large language models are not robust multiple choice selectors." arXiv preprint arXiv:2309.03882 (2023).
>
> > [2] Li, Jiatong, et al. "Perteval: Unveiling real knowledge capacity of llms with knowledge-invariant perturbations." Advances in Neural Information Processing Systems 37 (2024): 10679-10706.

---

### Official Review · Reviewer_mbsE · 2025-11-02

**Soundness:** 4
**Presentation:** 3
**Contribution:** 3
**Rating:** 6
**Confidence:** 5

**Summary:**

This paper investigates LLM benchmarking methodology.  It begins by exploring the fact that some LLMs can answer multiple-choice benchmarking questions without actually looking at the question, implying that they are influenced directly by the content of the answers, as opposed to a true understanding of the question. This suggests some natural measures of the influence of the choice text versus the question text.  A probabilistic view on these measures further suggest the NPSQ (for "Normalized Probability Shift for the Question") which normalizes the vanilla measure to account for different baselines.

Empirically, the paper explores the choice sensitivity and NPSQ on a small set of models and benchmark tasks.  Results show that a surprisingly high percentage of performance is attributable to choice sensitivity, suggesting both improved measures are needed (and, I will add, improved benchmark questions, although this is not argued for in the paper).

**Strengths:**

I liked this paper. I think it's well-written, addresses an interesting issue, primes other researchers for future work in this area, and makes non-obvious contributions to the literature.

* Understanding choice sensitivity seems like an important issue in benchmark design.

* The method of calculating choice sensitivity is natural and intuitive.

* The empirical results convincingly show a wide variety of surprising behavior of LLMs wrt choice sensitivity.

* The authors (generally) do a great job of writing, and highlighting important conclusions.

Despite its weaknesses, I recommend acceptance.

**Weaknesses:**

I think the biggest weakness of the paper lies in the presentation.

I think Section 3 was beautiful.  It flowed naturally, the experiments were clean, and the authors did a great job of pulling out crisp conclusions.

Section 4 was fine; it introduces NPSQ. It would have been nice to connect the mathematical notation in Section 3 to that in Section 4 a bit more directly -- it was VERY unclear what the "score" function in Section 3 was -- and since it was mentioned that "log p(x|q,c)" was part of it, it seems like there are some unstated notational overlaps between the sections.

Where things get muddled is Section 5.  I supposed I expected to see a clean comparison of NPSQ vs. the vanilla Choice Sensitivity in Section 3, but instead, the authors *also* introduced the idea of adversarial prompting.  This came out of nowhere, and (to me, at least) derailed the narrative flow.  I kind of see how it was designed to really skew the choice information, and therefore demonstrate how NPSQ was more robust the regular CS, but it was pretty unclear to me why this idea was introduced in Section 5 -- it seems like it should have been introduced earlier, or not at all.

I of course wish that the authors had tested on a wider variety of LLMs.  I understand computational limitations and all, but still - it seems that, as a benchmarking paper that is entirely empirical, more could have been done.

**Questions:**

* Why wasn't adversarial prompting introduced earlier (in sec 3?) and evaluated as part of the basic CS experiments?

* I was interested to see how choice sensitivity decreases as a function of model size. It seems like you could have tested on 70/80b variants of several of your models; is there a reason you didn't?

---

> ### Author Response · Authors · 2025-11-21
>
> We appreciate your valuable feedback. We address the individual comments below:
>
> Q1: The score function defined in Section 3 is not clearly connected to the formulas presented in Section 4, and it would strengthen the narrative to introduce adversarial prompting earlier.
>
> A1: We have revised Section 3 to discuss adversarial prompting and Section 4 to clarify that NPSQ connects to the score function defined in Section 3. Please see the paragraph of the updated version at line 296 and 342.
>
> Q2: The observation that choice sensitivity decreases with model size is interesting, and it would be valuable to include evaluations on 70B/80B variants as well.
>
> A2: We have added results for Qwen2.5 14B, 32B and 72B to address this concern. Please refer to Figure 1(a) in the updated version.

---

### Meta-Review · Area_Chair_g1qL · 2025-12-26

**Summary:**

Main contribution

The paper studies the “choice‑sensitivity” effect in Multiple Choice Question Answering (MCQA) evaluation; it derives a formal decomposition of a model’s log‑likelihood into choice‑driven and question‑driven components and proposes NPSQ (Normalized Probability Shift by the Question), a simple, auditable metric that compares the log‑likelihood of each option with and without the question and normalises the resulting shift.

According to the reviewers, the paper addresses a well‑known evaluation shortcut and provides an intuitive metric that can help understand if and when LLM capabilities are due to a deep understanding of the question, requires no training, and is easy to plug into existing evaluation pipelines. Robustness is well demonstrated via adversarial choice perturbations and format changes. Overall, the paper is well written and easy to follow.

Reviewers state that normalisation is not compared to plausible alternatives (z‑score, temperature scaling, ECE) and that the stability claims lack uncertainty bounds or statistical tests (authors argue their evaluation is deterministic due to fixed parameters). The proposed method still ignores permutation bias (option order) which may influence the choice component. Originally, experiments were limited to models ≤ 8 B, larger models are now added but not extensively discussed. Reviewers state that the evaluation cost is a non‑trivial overhead and that the contribution is an incremental methodological tweak rather than a fundamentally new approach, yet still provides meaningful insights.

**Reviewer Concerns:**

Addressed:
- Model size - experiments on models >8B are conducted, confirming that the basic trend holds and NPSQ remains effective
- Evaluation cost - this is a concern, but the authors put their method into perspective
- Plausible alternatives (partially addressed) - I understand the authors' reasoning, but I could imagine the reviewer to expect a more detailed analysis of this issue

Still Open
- Permutation bias - reviewers would be justified in asking this to be included in this analysis, I think
- Statistical tests - surely these could be added with additional overhead

**Reviewer Scores:**

Based on the authors' rebuttal, I expect reviewers to raise their scores to 7, maybe with the exception of reviewer WE2p who may want to see further ablations on permutation bias before raising their score.

---

### Decision · Program_Chairs · 2026-01-26

Accept (Poster)